# Design, Operation and Optimization of Constructed Wetland for Removal of Pollutant

**DOI:** 10.3390/ijerph17228339

**Published:** 2020-11-11

**Authors:** Md Ekhlasur Rahman, Mohd Izuan Effendi Bin Halmi, Mohd Yusoff Bin Abd Samad, Md Kamal Uddin, Khairil Mahmud, Mohd Yunus Abd Shukor, Siti Rozaimah Sheikh Abdullah, S M Shamsuzzaman

**Affiliations:** 1Department of Land Management, Faculty of Agriculture, Universiti Putra Malaysia, Serdang 43400, Malaysia; ekhlasurrahman02@gmail.com (M.E.R.); myusoffas@upm.edu.my (M.Y.B.A.S.); mkuddin@upm.edu.my (M.K.U.); 2Divisional Laboratory, Soil Resource Development Institute, Krishi Khamar Sarak, Farmgate, Dhaka-1215, Bangladesh; shamsuzzamansm@gmail.com; 3Department of Crop Science, Faculty of Agriculture, Universiti Putra Malaysia, Serdang 43400, Malaysia; khairilmahmud@upm.edu.my; 4Department of Biochemistry, Faculty of Biotechnology and Biomolecular Science, Universiti Putra Malaysia, Serdang 43400, Malaysia; mohdyunus@upm.edu.my; 5Department of Chemical & Process Engineering, Faculty of Engineering & Built Environment, Universiti Kebangsaan Malaysia, UKM Bangi 43600, Malaysia; rozaimah@ukm.edu.my

**Keywords:** constructed wetland, wastewater treatment, wetland plants, pollutant removal

## Abstract

Constructed wetlands (CWs) are affordable and reliable green technologies for the treatment of various types of wastewater. Compared to conventional treatment systems, CWs offer an environmentally friendly approach, are low cost, have fewer operational and maintenance requirements, and have a high potential for being applied in developing countries, particularly in small rural communities. However, the sustainable management and successful application of these systems remain a challenge. Therefore, after briefly providing basic information on wetlands and summarizing the classification and use of current CWs, this study aims to provide and inspire sustainable solutions for the performance and application of CWs by giving a comprehensive review of CWs’ application and the recent development of their sustainable design, operation, and optimization for wastewater treatment. To accomplish this objective, thee design and management parameters of CWs, including macrophyte species, media types, water level, hydraulic retention time (HRT), and hydraulic loading rate (HLR), are discussed. Besides these, future research on improving the stability and sustainability of CWs are highlighted. This article provides a tool for researchers and decision-makers for using CWs to treat wastewater in a particular area. This paper presents an aid for informed analysis, decision-making, and communication. The review indicates that major advances in the design, operation, and optimization of CWs have greatly increased contaminant removal efficiencies, and the sustainable application of this treatment system has also been improved.

## 1. Introduction

Water stress is now a reality in many parts of the world [1,2,3,4]. This phenomenon is destined to worsen considering that the consumption of water is expected to increase significantly in the coming years and with now more evident climate change accentuating this phenomenon [5]. In the European Union (EU), at least 17% of the territory and 11% of the population is affected by water scarcity [6,7,8,9,10,11]. In the Mediterranean area, over 20% of the population lives under constant water stress and in the summer this percentage reaches 50%, as defined by the European Environmental Agency’s (EEA) water exploitation index (WEI) [12,13,14].

In addition to reducing water consumption, a possible solution can be the reuse of treated wastewater (WW) produced by human activities [15,16,17]. The WW presents a very large number of contaminants such as dyes, surfactants, heavy metals, drugs, personal care products and bacteria [18,19,20,21,22,23]. To date, there are numerous treatments to remove the contaminants present in the WW, but there is still little attention to the possible reuse of treated water. For example, in the EU 40,000 million m^3^ of treated WW are produced every year, of which only 964 million m^3^ are reused [7]. Failure to recover the treated WW also involves developing countries where in many cases a significant plant shortage must also be addressed.

At present, there are growing issues of water environment including water shortage, water pollution and degradation of water resources worldwide. Moreover, the situation is becoming more serious due to the combined effects of worsening environmentally unfriendly activity and large population especially in developing countries [24,25]. Traditionally, conventional controlled wastewater management schemes have been effectively employed in many other regions to regulate water contamination [26]. Such wastewater management strategies, like the activated sludge cycle and membrane bioreactors, including membrane isolation, are therefore very costly moreover not completely viable for broad use in local communities [27]. In addition, these are constrained along with inadequate while dealing with ever greater requirements for water including wastewater management [28]. Particularly in evolving areas, choosing inexpensive and effective replacement techniques for wastewater purification is therefore important. Constructed wetlands (CWs), as a rational choice for managing pollutants, are drawing significant attention for this intention due to less expense, lower operational and management specifications [29]. Constructed wetlands (CWs) are green and engineered wastewater treatment systems, which are designed and constructed to utilize the natural purification processes involving wetland plants, substrates, and the associated microbes [30].

Constructed wetlands (CWLs) are engineered systems built for sewage treatment utilizing natural processes of soil, substrate, plant, and microorganism with a synergistic combination of physical, chemical, and biological functions [31,32] CWL systems have developed rapidly in the past three decades and been extensively employed to treat multifarious sewage, such as domestic sewage, industrial effluent, stormwater, polluted river water, and raw drinking water [33]. Owing to their comparatively low cost, easy operation, and maintenance, CWLs offer a reliable, suitable, and green treatment process for developed regions and economically underdeveloped areas. Although CWLs have been proven to be efficient for conventional and nonconventional pollutants, several intrinsic drawbacks, including substrate clogging, low pollutant removal efficiency, and disability for specific recalcitrant pollutants [25,33] limit their further application. Presently, lots of experiments have concentrated on the design, production, and efficiency of CWs and it has already been documented which CWs can be effective in eliminating different toxins from wastewater (organic matter, nutrients, trace elements, pharmaceutical substances, pathogens, and so on) [34,35]. Recently, CWLs coupled with other treatment technologies have been investigated to overcome the shortages of individual CWL systems, maximize the combined advantages, and construct a combined win-win system [36,37].

Conversely, successful long-term care success at CWs and reliable service represent a major problem. In the first position, plant varieties including media characteristics are critical variables in the elimination efficiency of CWs because they are regarded the key biological feature of CWs and alter the key elimination systems of contaminants directly or indirectly across period [38,39]. In the second position, the management efficiency of CWs is highly reliant on the optimum operational variables (water depth, hydraulic retention time along with load, feeding mode including setup configuration, and so on) that could occur in variability in the efficiency of elimination of pollutants between multiple experiments [25,40]. In comparison, a number of contaminant elimination methods (e.g., sedimentation, filtration, precipitation, volatilization, adsorption, plant uptake, including multiple biological systems) are typically directly and/or indirectly affected through multiple interior and exterior factors, like temperatures, the abundance of dissolved oxygen along with organic carbon sources, operating methods, and pH, including redox factors in CWs [35,41,42,43].

Whereas much progress has been made in the toxic compound elimination methods at CWs over the decades, there is indeed a difference in comprehending these mechanisms that is restricted to achieving constant amounts of enhancement in water quality. In the meantime, in current decades, the in-depth information reported in global journals and books about improving care efficiency has significantly enhanced. Furthermore, the latest progress and information on the feasibility of CW management method needs to be checked and addressed. The focus of this article is to characterize a wide range of CW methods and offer a general overview of how CWs have been applied to wastewater management in recent decades. This article further examines advancements in CWs that regard plants along with substrates to identify and optimize operational parameters for the stabilization of wastewater managements. Furthermore, subsequent study considerations are addressed for enhancing the stability of CWs.

## 2. Constructed Wetland 

Constructed wetlands (CWs) are designed and operated wetlands that are built to imitate unique ecological processes for wastewater treatment. These technologies, that consist principally of plants, substrates, soils, microbes and water, usage various tasks encompassing physical and chemical, including biological, strategies to eliminate multiple pollutants or strengthen the water freshness [24,35].

## 3. Classification of Constructed Wetland

Figure 1 demonstrates a basic structure for different forms of CWs. As shown in Figure 1, constructed wetlands are usually divided into 2 forms as per the wetland hydrologic processes: free water surface (FWS) CWs along with subsurface flow (SSF) CWs [35]. FWS structures are closer to natural wetlands where wastewater flows shallowly over polluted substrates. In SSF structures, wastewater passes horizontally or vertically across the substratum that encourages plant production, and can be also separated into vertical flow (VF) along with horizontal flow (HF) CWs on the ground of the stream route. A mixture of multiple wetland technologies is recognized as hybrid CWs, further incorporated for wastewater recovery, and this configuration is usually composed of two phases of multiple model parallel CWs, like VF-HF CWs, HF-VF CWs, and HF-FWS CWs, as well as FWS-HF CWs [44]. Additionally, multiple stage CWs consisting of more than 3 CW steps are utilized [40]. Augmented CWs like manmade ventilated CWs, baffled flow CWs, hybrid tower CWs, step feeding CWs together with circular flow corridor CWs have been introduced in current years to progress the efficiency of wastewater recycling processes [25].

## 4. Utilization of CWs for Wastewater Treatment

The implementation of CWs is largely utilized quickly in the treatment of conventional domestic as well as municipal wastewater. Currently, the use of CWs has been enlarged noticeably to cleanse farmland pollutants, industrial wastewater, mine storm water drains, landfill leachates, contaminated river including lake waters, including metropolitan along with road runoff, and has even been established under different environmental situations like hot along with humid environment, arid including cool climate, tropical climate globally [25]. Furthermore, CWs are a viable solution for hazardous waste management in emerging countries, and large number of CWs have been implemented as wastewater management utilities particularly in China [42]. Cocopeat, zeolite, as well as limestone wetlands experienced strong As elimination levels (over 98% on average) for the whole research span, while levels declined over duration for gravel wetlands, either as a percentage and as a daily mass extracted per capacity.

According to National Water Commission (Comisión Nacional del Agua - CONAGUA) [45], of the municipal collected wastewater in Mexico, only 63% is treated, which shows the clear need for wastewater treatment plants; however, the implementation of this is not a common case due the high costs required for the construction and operation thereof. Therefore, the use of ecologically viable or sustainable alternatives to solve these problems is needed; for this purpose, constructed wetlands (CWs), or so-called artificial wetlands are examples of such an alternative [32,46,47]. Improving water quality is a relevant environmental aspect, and using constructed wetlands (CWs) is a sustainable option for this. Both porous materials filled cells and plants that collectively remove contaminants must be readily available and inexpensive. This study evaluated CWs and their functionality by comparing two ornamental plants (*Spathiphyllum wallisii* and *Hedychium coronarium*) planted in experimental mesocosm units filled with layers of porous river rock, tepezil, and soil, or in mesocosms with layers of porous river rock, and tepezil, without the presence of soil. The findings during the experiments (180 days), showed that the removal of pollutants (chemical oxygen demand (COD), total solids suspended (TSS), nitrogen as ammonium (N-NH_4_), as nitrate (N-NO_3_), and phosphate (P-PO_4_) was 20–50% higher in mesocosms with vegetation that in the absence of this, and those mesocosms with the soil layer between 33–45% favored removal of P-PO_4_. Differences regarding of vegetation removal were only observed for N-NH_4_, being 25–45% higher in CWs with *H. coronarium*, compared with *S. wallisii* [48].

Moreover, for the appropriate wetland media, the discharge of Arsenic (As) in relatively large quantities was more or less the same, but it only significantly reduced whenever the density of inflows reduced [49]. During this experiment, gravel, zeolite (microporous aluminosilicate mineral), ceramsite (light weight enlarged clay accumulation) along with manganese sand were measured as potential substrates whilst aquatic *Juncus effuses* (Soft Rush or Common Rush) as well as terrestrial *Pteris vittata L.* (Chinese Ladder Brake; identified as As hyperaccumulator) were verified as prospective wetlands plants. The findings of batch deposition demonstrated that manganese sand had a highest As(V) adsorption level of 4.55 h^−1^ as well as an uptake ability of 42.37 μg g^-1^ as comparison to the numerous different 3 aggregate particles [25]. Contrary to manganese sand covered wetlands, the existence of *J. effuses* including *P. vittata* result in an enhanced mean elimination of As(V) by substantially 21 along with 10% separately for wetlands A including B relative to uncultivated wetland E. In addition, As(V) degradation efficiency was significantly affected with meterological temperature variations. The maximum and minimum As(V) extraction efficiency of 83% and 43% for Wetland A took place in hot September along with a cold January, separately [25]. FWS CWs are often more powerful in removing organics including suspended solids than FWS CWs and SSF CWs, comparison to removing nitrogen as well as phosphorus [40]. SSF CWs are extremely competent in extracting organics, suspended solids, microbiological contamination, and HMs contrast to FWS CWs, and are less susceptible to cold and simpler to isolate for winter performance.

The current knowledge about the role terrestrial ornamental plants play in constructed wetlands (CWs) has scarcely been evaluated. Likewise, little attention has been given towards the use of new support or fill media for subsurface flow CWs, which may result in the reduction of costs when implemented on a large scale. This study evaluated, during nine months, the effect of three terrestrial ornamental plants and two substrates on the elimination of pollutants in wastewaters by using fill-and-drain vertical subsurface flow CWs (FD-CWs). Sixteen microcosms were used, nine filled with polyethylene terephthalate (PET) and nine with porous river stone (PRS). For each type of substrate, duplicates of microcosms were used, utilizing *Anthurium* sp., *Zantedeschia aethiopica*, and *Spathiphyllum wallisii* as vegetation and two other CWs without vegetation as controls. The environmental conditions, number of flowers, and height of the plants were registered. The results revealed that both substrates in the FD-CWs were efficient in removing pollutants. The average removal of pollutants in systems with vegetation revealed a positive effect on the reduction of the biochemical oxygen demand (55–70%), nitrates (28–44%), phosphates (25–45%), and fecal coliforms (52–65%). Meanwhile, in units without vegetation, the reduction of pollutants was nearly 40–50% less than in those with vegetation. The use of PET as a filling substrate in CWs did not affect the growth and/or the flowering of the species [50].

The vegetation in constructed wetlands (CWs) plays an important role in wastewater treatment. Popularly, the common emergent plants in CWs have been vegetation of natural wetlands. However, there are ornamental flowering plants that have some physiological characteristics similar to the plants of natural wetlands that can stimulate the removal of pollutants in wastewater treatments; such importance in CWs is described here. A literature survey of 87 CWs from 21 countries showed that the four most commonly used flowering ornamental vegetation genera were *Canna, Iris, Heliconia,* and *Zantedeschia*. In terms of geographical location, *Canna* spp. is commonly found in Asia, *Zantedeschia* spp. is frequent in Mexico (a country in North America), *Iris* is most commonly used in Asia, Europe, and North America, and species of the *Heliconia* genus are commonly used in Asia and parts of the Americas (Mexico, Central, and South America). This review also compares the use of ornamental plants versus natural wetland plants and systems without plants for removing pollutants (organic matter, nitrogen, nitrogen and phosphorous compounds). The removal efficiency was similar between flowering ornamental and natural wetland plants. However, pollutant removal was better when using ornamental plants than in unplanted CWs. The use of ornamental flowering plants in CWs is an excellent option, and efforts should be made to increase the adoption of these system types and use them in domiciliary, rural and urban areas [51].

Pharmaceutical and personal care products (PPCPs) are chemicals employed in human healthcare, veterinary, medical diagnosis, and cosmetics, which have increasingly polluted water sources. Extensive research has demonstrated constructed wetlands (CWs) technology as a low-cost but efficient approach for PPCPs removal. There has been a growing interest to better understand the degradation mechanisms of PPCPs in wetland cells. Data corroborated in this review show that these degradation mechanisms include photolytic degradation, adsorption, phytodegradation, and microbial degradation. Each of these degradation mechanisms performs differently in wetland cells [52].

## 5. Factors Influencing on Design of Constructed Wetland

Phytoremediation has been investigated systematically in constructed wetland around the globe. The productivity of the constructed wetland was based on the differential ecological variables, the various behaviour of the plants and their related pollutant rhizobacteria. The structure and densities of pollutants existing, soil/air moisture encompassing, pH, temperature, dissolved oxygen (DO), soil elementary material and solubility, and facilitating microbial media are influenced by the level of adsorption during phytoremediation [53].

Constructed wetlands (CWs) were comprehensively investigated for the treatment of pollutant from wastewater. As summarized in Table 1, most of the studies considered one or more topics for evaluation, such as role of design and/or operational factors, role of physicochemical parameters, effect of plants and/or support matrix, and impact of seasonality (summer and winter) on the removal of pollutant in CWs.

### 5.1. pH

The development of plants as well as nitrification and heterotrophic microbial activities in Constructed wetlands (CWs) requires optimal pH values (near neutral) [58,62]. The pH of the influent wastewater controls several biotic processes [58] and the degree of ionization of the ionizable compounds [63], therefore, it can be considered an important parameter. The presence of plants in CWs regulates the pH (~7.5) and influences the treatment performance [64]. The high effluent pH affects the adsorption behavior of Emerging organic contaminants (EOCs) due to their dissociation and subsequent attachment to soil/sediment by ion exchange [65]. Additionally, the pH of the system is positively correlated with DO, which enhances the removal of personal care products (PCPs) for which aerobic conditions are more favorable.

### 5.2. Temperature

Temperature exhibited a positive correlation with the removal efficiency of five of the six studied PCPs, although significant positive correlation with the removal efficiency of galaxolide, methyl dihydrojasmonate, and tonalide. Some studies revealed that microbial degradation is their possible removal pathway in CWs, which is enhanced at warm temperature (15–25 °C), particularly in the case of nitrifying and proteolytic bacteria [54,58,61,62].

### 5.3. Dissolved Oxygen (DO)

The effluent dissolved oxygen (DO) exhibits a positive correlation with the removal efficiency of five of the six studied PCPs, although a significant positive correlation is with the removal efficiency of galaxolide, tonalide, and triclosan, which represents the importance of DO in the removal processes of these PCPs as noted by several studies (e.g., References [55,57,66]. For instance, Ávila et al., [55] attributed the enhanced removal efficiency of tonalide (83%) in aerated (AA) vertical flow constructed wetland (VFCW) VFCW compared with non-aerated (NA) VFCW (61%) to the elevated level of DO in AA-VFCW compared with NA-VFCW (5.2 and 3.4 mg L^−1^, respectively). Similarly, the authors observed the better removal efficiency of oxybenzone and triclosan in AA-VFCW (91% and 86%, respectively) compared with NA-VFCW (89% and 73%, respectively). The available evidence suggests that aerobic biodegradation is one of the major removal mechanisms of methyl dihydrojasmonate, triclosan, and oxybenzone in CWs [52].

## 6. Constructed Wetlands in Developing Countries

Inadequate access to clean water and sanitation has become one of the most pervasive problems affecting human health in developing countries, and problems with water are expected to worsen in coming decades [67,68]. Developing countries are defined according to their gross national income (GNI) per capita per year. Countries with a GNI of US$ 12,615 and less are defined as developing [69]. A list of developing countries is provided in Table 2 (According to the World Bank, [69]).

According to a recent report by World Health Organization [70], more than one-tenth of the global population (780 million) still relied on sub-standard drinking water sources in 2010. Lack of sanitation is an even larger concern. An estimated 2.5 billion people are still without improved sanitation, and sanitation coverage is below 50% in many countries of Sub-Saharan Africa and Southern Asia. Consequently, millions of people die annually (3900 children per day) from diseases transmitted through unsafe water or human excreta [68]. With insufficient water resources to meet rising water demand, many sources of water (e.g., groundwater) that are considered easy to be developed geographically and technologically have been overexploited in many developing countries [71,72]. This short-term strategy is likely to have detrimental effects on the environment, such as ground subsidence, salinity intrusion, and ecosystem deterioration [73]. In addition, many cities in developing countries have also generally fallen behind in constructing and managing sewage treatment facilities. Among the various developments, treatment of wastewater is always considered one of the lowest priorities [74]. The consequence of this is the common practice of discharging large amounts of untreated wastewater directly into streams and lakes in many developing countries [75,76] Ecological technologies such as constructed wetlands for wastewater treatment represent innovative and emerging solutions for environmental protection and restoration, placing them in the overall context of the need for low-cost and sustainable wastewater treatment systems in developing countries [24,77].

CWs have always been a famous wastewater management option over the past several years, and have been accepted as effective approaches to traditional wastewater management processes. This is due to the increased effectiveness in removing pollutants, simple construction and operation, lower power requirements, high wastewater management rates and the possibility to provide substantial natural habitats [24]. In emerging and underdeveloped countries, while CWs have been utilized to manage domestic wastewater [78], the usage of CWs has also been progressively prolonged to other forms of wastewater like manufacturing wastewater [79], farm wastewater [80], lake/river water [39,81], Sludge sewage [82], wastewater formed by oil [83], storm-water overflow [84], wastewater of sugar industry [85], hospital wastewater, lab wastewater, landfill leachate [86], together with farm overflow [80,87].

## 7. Design and Operation of Constructed Wetland

The characteristics for design as well as implementation of the Constructed Wetland (CW) contained site choice, choice of plants, choice of substrates, category of wastewater, choice of plant substances, hydraulic loading rate (HLR), Hydraulic retention time (HRT), depth of water, mood of construction along with conservation processes [40,88]. Figure 2 provides the outline of a constructed wetland. Specifically, factors like plant quality, substratum quality, water depth, hydraulic loading rate (HLR), hydraulic retention time (HRT), along with feeding mood may be critical to creating a successful CW process, including achieving successful care.

### 7.1. Plant Choice in Constructed Wetlands

Just a few of plant varieties were commonly utilized in developed wetlands [89]. Thus, the selection of plants utilized in CWs should be the emphasis of the ongoing study on green CW layout [24]. In addition to adapting to severe weather situations, the resistance of waterlogged-anoxic as well as hyper-eutrophic situations, including capability of uptake of pollutants, are suggested for plant choice.

#### 7.1.1. Plants Utilized in Constructed Wetlands

Regularly utilized macrophytes in CW managements involve developing plants, waterlogged plants, floating leaved plants including free-floating plants. While over one hundred fifty varieties of macrophytes have already been utilized worldwide in CWs, in fact mostly a small number of such varieties of plants are often most frequently cultivated in CWs [89]. The another very widely utilized evolving plant varieties are *Phragmites* spp. (Poaceae), *Typha* spp. (Typhaceae), *Scirpus* spp. (Cyperaceae), *Iris* spp. (Iridaceae), *Juncus* spp. (Juncaceae) including *Eleocharis* spp. (Spikerush). *Hydrilla verticillata, Ceratophyllum demersum, Vallisneria natans, Myriophyllum verticillatum* as well as *Potamogeton crispus* are the most commonly prescribed waterlogged plants. The floating leaved plants are usually *Nymphaea tetragona, Nymphoides peltata, Trapa bispinosa* as well as *Marsilea quadrifolia*. The free-floating plants are *Eichhornia crassipes, Salvinia natans, Hydrocharis dubia* together with *Lemna minor.*

Amongst the aforementioned macrophytes, rapidly growing plants are the major vegetation in FWS as well as SSF CWs engineered for wastewater managements. Vymazal, [89] studied emerging plants utilized in FWS CWs and found that *Phragmites australis* is the very common varieties in Europe as well as Asia, *Typha latifolia* in North America, *Cyperus papyrus* in Africa, *P. australis* including *Typha domingensis* in Central / South America along with *Scirpus validus* in Oceania, respectively. In the same way, Vymazal’s [90] investigation on plants utilized for the SSF CWs exhibited that perhaps the very frequently utilized plant in the world is *P. australis*, that has been utilized in specific across the whole of Europe, Canada, Australia, and many other areas of Asia as well as Africa. *Typha* spp. (for example, *latifolia, domingensis, orientalis* including *glauca*) are the 2nd most frequently utilized SSF CW variety, and they are utmost popular in North America, Australia, Africa including East Asia. Scirpus (for example, *lacustris, Validus, acutus* as well as *californicus*) are many widely found plants that are mainly distributed in North America, Australia along with New Zealand. *Juncus effusus* including *Eleocharis* sp. can be attributed widely in Asia, Europe, and North America [24]. In addition, a few other decorative variety (like *Iris pseudacorus*) are utilized for CWs particularly in the tropical as well as subtropical areas [91].

#### 7.1.2. Plant Resistance against Wastewater

Wetland plants would possibly encounter from ecological discomfort whenever CW interventions are utilized to eliminate specific contaminants. Surrency, [92] observed that the severe wastewater environments could surpass plant resistance and reduce the capacity for plant survival as well as management. Especially whenever confronting heavy wastewater levels or purifying wastewater that contains harmful contaminants, CW managements could scarcely perform efficiently due to decreased plant survival [92]. Environmental changes may also trigger significant disruption to wetland plants; for instance, eutrophication would impede plant development including even trigger plant loss. Xu et al., [93] further noted that substantial quantities of ammonia can harm photosynthetic activity and minimize plant nutrient absorption. Additional ammonia may affect chlorosis in the leaves, inhibit growth, lead to smaller root and yield downturns in visible signs, and induce oxidative stress as demonstrated by catalase and peroxidase enhancements [93]. Based on the aforementioned evidence, a variety of study have been published out to assess the capability of resistance to pollutant rates of different wastewater. Chen et al., [94] noticed that the *Typha angustata* could persist at greater levels up to 30 ppm of Cr (VI) solution for 20 days and had an outstanding aggregation capability. *Arundo donax* along with *Sarcocornia fruticosa* have the capacity to cure strong salinity wastewater (up to 6.6 g Cl L^-1^), and to be extremely successful in eliminating organics, nitrogen including phosphorus [95]. Therefore, these studies are necessary not only for determining the resistance of wetland plants, but also for choosing the very resistant plant varieties in CW wastewater managements.

#### 7.1.3. Survey Results of the Use of Ornamental Flowering Plants in CWs

Many CWs around the world used OFP for the removal of various types of wastewater (Table 3). For example, in China, the most popular plants used is *Canna* sp., while in Mexico the ornamental plant used is more diverse, including plants with flowers of different colors, shapes, and aromatic characteristics (*Canna, Heliconia, Zantedeschia, Strelitzia* spp.). 

A review of the available literature showed that ornamental plants are used to remove pollutants from domestic, municipal, aquaculture ponds, industrial or farm wastewater. The removal efficiency of ornamental plants was also evaluated for the following parameters: biochemical oxygen demand (BOD), chemical oxygen demand (COD), total suspended solids (TSS), total nitrogen (TN), total phosphorous (TP), ammonium (NH_4_-N), nitrates (NO_3_-N), coliforms and some metals (Cu, Zn, Ni and Al). There is no clear pattern in the use of certain species of ornamental plants for certain types of wastewater. However, it is important to keep in mind that CWs using ornamental plants are usually utilized as secondary or tertiary treatments, due to the reported toxic effects that high organic/inorganic loading has on plants in systems that use them for primary treatment (in the absence of other complementary treatment options) [179,180]. The use of OFP in CWs generates an esthetic appearance in the systems. In CWs with high plant production, OFP harvesting can be an economic entity for CW operators, providing social and economic benefits, such as the improvement of system landscapes and a better habitat quality. Some authors have reported that polyculture systems enhanced the CW resistance to environmental stress and disease [99,174].

#### 7.1.4. Common Ornamental Plants Used in CWs

Limited quantities of ornamental flowering plants (OFP) have been used in CWs. These types of plants are typical of subtropical and tropical regions. Our survey showed that the four most frequently used genera are, in order of most to least frequently used: *Canna* spp., *Iris* spp., *Heliconia* spp., *Zantedeschia* spp. (Table 4). Species of the Canna genus are used in all continents, with Asia using them the most frequently. The Iris genus is also used in Asia, along with Europe and North America. Species of the Heliconia genus are commonly used in Asia and America, including Mexico, Central and South America. While Zantedeschia is most frequently used in Mexico (a country in North America), they are found with less frequency in Europe, Africa, and Central and South America. The use of OFP in CWs is most popular in tropical and subtropical regions, due to the warm temperatures and the extensive sunlight hours. Such environmental features stimulate a richer biodiversity than in other regions.

#### 7.1.5. The Importance of Plants in CWs Environment

In CWs, the evolving biomass also recognized as the aquatic macrophytes performs a key function in ecological relationships with CWs through pollutant extraction and wastewater purification. For the effective metabolizing microbiological community in CW- ecosystems, the components of the aquatic microphytes i.e. the aerial parts like the roots, stalks including leaves serve as substrates and connection sites. In SSF-CWs, especially HSSF-CWs, in which the level of oxygen is typically considerably low, restricting essential oxidative functions such as nitrification along with nutrient bio-transformations [181], the roots of the emerging biomass move oxygen by diffusion to the root areas/rhizomes. Earlier experiments have recorded that CW plants can pass 5–45 g O_2_ d^-1^ m^-2^ depend on oxygen distress rate and plant intensity [181,182]. CW plants provide defence by coating toward freeze in winter season, controlling substrates blockage particularly in VSSF-CW schemes, ensuring surface bed stability whereas supplying optimum condition for many physical processes like filtration, regulating hydrodynamic properties like CWs flow rate, serving as wildlife and supplying CW-system attractiveness. CW plant choice also takes into consideration many variables in particular; plant varieties that can display strong resilience within severe climatic situations and show great resistance to eutrophic, acidic, hypoxic, and water-logged situations [32]. Plant effluent treatment techniques for pollutants sometimes contain; plant absorption, bio immobilization, and biosorption of substrates [183]. The evolving vegetation further shows role of phytoremediation (botanical bioremediation) to purify HMs from CW-ecosystems. Elsewhere, plants utilize several methods such as: (i) Phytoextraction (Gao et al., [113]) including the utilize of plants that can absorb the HM in their shooting cells to remove large amounts of HM from wastewater. Phytoextraction eliminates the excessive levels of Pb, Cd, Ni, Cu, Cr including Se. The shoot biomass is typically collected at a specific location for appropriate processing, or is burned to remove the metals. Because an in-situ combustion negatively impacts the CW habitats including raises the likelihood of reloading HM. Implementing an ex-situ disinfection phase is always a viable option, but this greatly raises the operating expenses and infringes a conventional CW system’s prospective cost-effectiveness. (ii) Phytostabilization [184,185,186] includes reducing the flexibility and solubility of HMs by CW plants uptake and deposition. (iii) Phytovolatilization [187] that mainly includes the absorption and discharge of unstable HM elements towards the atmosphere. This indicates strong performance for phytodepuration in the elimination of mercury (Hg) as well as As from wastewater. (iv) Phyto / rhizofiltration [188,189] includes plant usage for the absorption, accumulation, or precipitation of metals from soluble garbage. Now this adventitious root systems of the CW plants have a diverse variety of HM uptake with large surface area. The popular and widely utilized varieties of emerging new aquatic plants for wastewater purification systems are *Scirpus* spp. (Bulrush), *Phragmites* spp. (Common reed), *Typha* spp. (Cattail), *Juncus* spp. (Rush), *Eleocharis* spp. (Spikerush), *Iris* spp. (Iridaceae) as well as *Carex* spp. (Sedge) [2,32,182,183,190,191]). *Phragmites australis* including *Typha* spp. have provided the broader implementation in CW-wastewater management methods based on flood resistance, reproductive capacity along with prolific nature [192]. A latest research identified the macrophyte, *Typha angustifolia* as a possible hyper-accumulator for the elimination of Co, Cu as well as Pb from an FWS-CW using a controlled method to phytoremediation [193]. Meanwhile, the effectiveness of several other CW plants has currently been recorded, such as *Zantedeschia aethiopica, Cyperus alternifolius, Heliconia burleana, Canna indica, Acorus calamus,* and *Ipomoea aquatic* [194,195], including a waterlogged plant, *Elodea densa,* that has already demonstrated considerable capability in pollutant purification [196].

### 7.2. Substrate Choice in Constructed Wetlands

Particularly in CWs and SSF CWs, the substratum is the significant layout parameter, since it can include an acceptable growth medium for plants and also enable efficient wastewater mobility [40]. Additionally, substratum sorption can perform a major part in removing different contaminants like phosphorus [197]. A major problem is the choice of appropriate substrates for the usage in CWs for industrial wastewater management.

#### 7.2.1. Substrates Utilized for Constructed Wetlands

Substrate choice is assessed on the basis of the hydraulic conductivity and the absorption potential of contaminants. Weak hydraulic conductivity would occur in major blockage, significantly reducing process capacity, and low substrate biosorption may also impact CW ‘s long-term extraction efficiency [198]. As seen in Table 5, numerous experiments have been performed on the choice of wetland substrates, in specific for effective phosphorus elimination from wastewater, and the commonly utilized substrates involve predominantly natural substance, artificial media along with industrial by-products like gravel, sand, clay, calcite, marble, vermiculite, slag, fly ash, bentonite, dolomite, calcite, stone, zeolite, willastonite, activated carbon, light weight aggregates [35,91,199]. Findings from these experiments further indicate that substrates like sand, gravel as well as rock are the weak applicant for long-term phosphorus treatment, but through comparison, strong hydraulic permeability and phosphorus dissolution capability artificial and industrial products may be alternate substrates in CWs. Other models have proposed some documentation on substrate selection to maximize the extraction of nitrogen and organic substances, and the introduction of substrates like alum sludge, peat, maerl, compost and rice husk [35]. In addition, a combination of substrates (sand and dolomite) was added to phosphate elimination in CWs [200], and the combined substrates (substrate gravel, vermiculite, ceramsite along with calcium silicate hydrate) were often utilized in CWs to manage low nutrient content surface water [201]. Not only do these combined substrates have responsive surfaces for microbiological binding, they may also have a strong hydraulic conductivity to prevent short circuiting in CW.

#### 7.2.2. Sorption Capability of Substrates

Substrates can eliminate contaminants from wastewater by exchange, adsorption, precipitation together with complex formation. The adsorption ability of substrates varies along with their sorption ability may rely mainly on the substrate material, and may also be affected by hydraulic along with contaminant loading [227]. Xu et al., [205] analysed nine substrates’ phosphorus sorption efficiency and found that sand sorption efficiency varied from 0.13 g kg^-1^ to 0.29 g kg^-1^. Similarly, Huang et al., [228] documented the adsorption level of various ammonium extraction substrates in CWs, and their analysis revealed that the measured maximal adsorption of zeolite ammonium (11.6 g kg^-1^) was considerably greater than that of volcanic rock (0.21 g kg^-1^). Additionally, other studies tested the adsorption potential of a combination of various substrates utilized in CWs. In the VF CWs examined by Prochaska and Zouboulis [200], the phosphorus aggregation of a combination of river sand and dolomite (10:1, w/w) substrates was observed to be in the range of 6.5–18%, and the approximate high absorption potential of the sand and dolomite combination was 124 ppm P.

### 7.3. Factors Influencing the Optimization of Design and Operation

Site choice, plant choice, substratum choice, wastewater sort, plant material quality, hydraulic loading rate (HLR), hydraulic retention time (HRT), water depth, operating mood along with maintenance protocols [40,87] were the requirements for design and operation of the CW. Specifically, variables like plant choice, choice of substrates, water depth, hydraulic loading rate (HLR), hydraulic retention time (HRT) along with feeding mood may be key to facilitate a successful CW process. Calheiros et al., [95] checked horizontal subsurface flow CWs in a leather industry in Portugal for the cleaning of strong salt content tannery wastewaters (2.2–6.6 g Cl L^-1^). The researchers reported two wetlands cultivated with *Arundo donax* along with *Sarcocornia fructicosa* (each of which has surface area of 72 m^2^ as well as depth of 0.35 m). Only at 6 cm d^-1^ did the hydraulic loading rate of both wetlands act consistently for COD (65% elimination), BOD (73%), TSS (65%), NH_4_-N (73%), as well as TKN (75%) though the discharge of TP was marginally greater in *Arundo* wetland (83%) comparison to *Sarcocornia* wetland (79%). The process had the capacity to meet the elimination requirements. Kaseva and Mbuligwe, [229] utilized a pilot HF CW to extract chromium and turbidity from wastewater from a tannery in Tanzania. The HF wetland was loaded with 4–30 mm particle size crushed pumice and calcareous soil, as well as cultivated with *P. mauritianus*. The HLR mean was 10 cm d^-1^, and the HRT mean was 1.6 days. The inflow Cr density of 372 ppm in a cultivated wetland was decreased by 99.8%, whereas the elimination of an uncultivated control cell was marginally lesser by 92.5%. The turbidity decreases in cultivated as well as uncultivated cells contributed to 71% and 66% separately.

In Bangladesh Saeed et al., [35] investigated a combination VF-HF-VF constructed wetland pilot-scale system for tannery wastewater. The wetland systems were packed with substances accessible regionally: organic coco-peat (1st VF), cupola slag, a cast iron melting method (HF) including pea gravel (2nd VF) by-product and all systems were cultivated with *P. australis*. Densities of inflow were exceptionally severe: COD 11, 500 ppm, BOD: 4200 ppm, TSS 27,600 ppm, PO_4_: 30 ppm, NO_3_: 66 ppm as well as NH_4_: 111 ppm. The total performance of care was 98%, 98%, 55%, 87%, 50%, and 86%, separately facing extremely heavy loads (690 g COD m^-2^ d^-1^). The contaminants were slowly eliminated in all systems by the process apart for ammonia that was eliminated mainly in VF systems.

#### 7.3.1. Monitoring of Physical and Chemical Parameters 

The phytotoxicity study for all treatments reported by physiochemical variables like moisture, temperature (T), dissolved oxygen (DO), oxidation-reduction potential (ORP), pH, along with chemical oxygen demand (COD), biochemical oxygen demand (BOD), total suspended solids (TSS), total dissolved solids (TDS), benzene, toluene, ethylbenzene, xylenes, (BTEX), and so on. The tests usually demonstrated that the moisture varied from 37.3–0.4% to 41.6–3.3%. The median pH varied from 7.2 ± 0.1 to 8.0 ± 0.2 [230], that are the ordinary development range of plants [231]. Mean temperature levels varied from 25 °C to 28 °C in the spiked sand over the average 42 days for a tropical area [230]. The situations in a reed bed model can be differentiated as per Szogi et al., [232] with regard to whether it is aerobic or anaerobic by DO and ORP measured data. Along with [233,234] the interaction between do as well as ORP is linear, meaning ORP also enhances as DO enhances. The DO rates vary from 5.8 ± 0.1 to 7.9 ± 0.0 ppm [230]. As the ORP reached −170 mV, 0 ppm DO densities happened [233]. The negative ORP indicates the machine is in a reduction condition as per Akkajit and Tongcumpou, [235]. The ORP varies between −50 including −130 mV during anoxic situations [236]. The findings of this analysis indicated that the treatment condition was between aerobic as well as anoxic, with both the ORP varying from −8.2 ± 0.9 to −67 ± 0.1 mV [230]. Alterations in ORP as well as pH can major effect on the As species that present in the soil solution [237]. According to Valverde et al., [238] As(V) is more prevalent in soil involving a lot of oxygen, and according to Peshut et al., [239], the inorganic As speciation that influence the harmful effect of As. The harmful consequence of As on *L. octovalvis* enhanced while bioavailable As(V) enhanced, even though plants could accumulate more As(V). As absorption and bioaccumulation by plant continues the pattern As(V) > As(III)>monomethyl arsonic acid (MMA)>dimethyl arsinic acid (DMA) as per the ATSDR, [240]. It inverts the correlation between do as well as COD [241], that is, the smaller the DO, the greater the COD level. The statistical investigation of the physiochemical results indicated no significant variance in moisture content (*p* > 0.05). Conversely, parameters for pH, temperature, ORP, DO, including COD, displayed vast variation within the period (*p* < 0.05) [230].

#### 7.3.2. Water Depth

Depth of water has been a key variable in deciding the plant varieties are to be formed including it also affects the biochemical reactions accountable for eliminating pollutants through influencing the redox state along with dissolved oxygen rate in CWs [242]. Dwire et al., [243] investigated the relationship between water depth as well as dispersion of plant types in two riparian meadows in northeastern Oregon, United States of America. Their findings showed that abundance of species like wetland sedges was closely correlated with depth of the water table. In addition, [244] research, through contrasting 0.27 m deep wetland beds with 0.5 m deep, demonstrated that variations in contaminant transformations exist within structures of various depths. Correspondingly, García et al., [245] measured the impact of depth of water on the elimination of chosen pollutants in Horizontal Flow Constructed Wetlands (HF CWs) throughout the time of three (3) years. The findings showed that beds with a water depth of 0.27 m reduced stronger chemical oxygen demand, biochemical oxygen demand, ammonia including dissolved reactive phosphorus. Additionally, studies conducted by Aguirre et al. to evaluate the impact of water depth on the effectiveness of organic matter discharge in HF CWs stated that the comparative impact of various metabolic pathways differed with the depth of water.

#### 7.3.3. Hydraulic Load and Retention Time

Hydrology was one of the main variables in regulating wetland activities, and to obtain a suitable treatment efficiency, flow rate should also be controlled [246]. The optimum configuration of hydraulic loading rate (HLR) as well as hydraulic retention time (HRT) performed a major part in the effectiveness of elimination of CWs. Larger HLR encouraged faster transit of wastewater by the media, thereby decreasing the optimal time of communication. On the opposite, a suitable microbiological population in CWs could be formed and have enough contact time to eliminate pollutants at a lengthy HRT [35,91]. Huang et al., [247] recorded a drastic decline in ammonium and TN levels in industrial effluents, with increased HRT in domestic wastewater treatment CWs. Likewise, Toet et al., [248] observed successful elimination of nitrogen in CWs with an HRT of 0.8-day in comparison with the 0.3-day residence period test. A shorter HRT in CWs can be correlated with insufficient wastewater denitrification, and it is stated that the extraction of nitrogen involves a lengthy HRT comparison to that needed for organic extraction [246]. In addition, the impact of HRT can vary between CWs reliant on the predominant plant varieties and temperature, as those variables directly influence wetland hydraulic performance. Correspondingly, a slight reduction in ammonium including TN extraction from domestic wastewater in Vertical Flow Constructed wetlands (VF CWs) was reported in a long-term research by [34], while HLR shifted from 7 to 21 cm d^-1^. Mean ammonium elimination therefore declined from 65% to 60%, while TN declined from 30% to 20%. Nonetheless, Stefanakis and Tsihrintzis, [249] published a long-term evaluation of fully developed VF CWs for the treatment of synthetic wastewater, and found that greater nitrogen and organic extraction was gained by wetland processes as the HLR rose. Avila et al. further analyzed the effectiveness of hybrid CW schemes utilized to remove developing organic pollutants, and found that the efficacy of elimination for many other substances declined as the HLR enhanced.

#### 7.3.4. Feeding Mode of Influent

Some other key design parameter has already been demonstrated to be the feeding mode of influent [250]. The variation in feeding mode (like continuous, batch including intermittent) can affect the situations of oxidation-reduction along with oxygen transfer as well as diffusion in wetland schemes and thus alter the efficacy of treatment. Several trials have been performed to determine the influence of powerful feeding modes on the efficacy of CW managements in elimination. By encouraging more oxidized situations, batch feeding mode may generally get higher outcome than continuous action. Zhang et al., [251] studied the impacts batch versus continuous flow on the efficiency including productivity of elimination in tropical SSF CWs. They stated that wetlands with batch flow mode demonstrated substantially greater efficiency gains in ammonium elimination (95.2%) comparison to continuously fed (80.4%) schemes. Fortunately, doubt persisted about whether batch operation enhanced extraction efficiency comparison with continuous feeding mode. Intermittent feeding mode could be regarded to improve the elimination matter and nitrogen in CWs [35]. Caselles-Osorio and García, [252] assessed the impact of continuous as well as intermittent feeding modes on the efficacy of pollutant elimination in SSF CWs and observed that intermittent feeding increased the efficacy of ammonium extraction in wetland schemes comparison to continuous feeding. In the continually fed systems, therefore, sulfate elimination was larger comparison with the intermittently fed systems. Jia et al., [253] also investigated the impacts of intermittent operation including various length of drying time on extraction efficiency in VF CWs, and comparison to continuous operation in wetland schemes, the intermittent operation encouraged a slower rate of COD including TP extraction. In addition, the intermittent operation significantly improved the performance of ammonium extraction (over 90%), that could be due to more oxidizing situations in wetlands. Equally, Jia et al., [254] estimated the impact of continuous as well as intermittent feeding modes on the extraction of nitrogen in FWS along with SSF CWs. Studies revealed that the intermittent feeding mode significantly augmented ammonium extraction in SSF CWs, with no noticeable impact on FWS CWs.

## 8. Performance and Efficiency Assessment of Constructed Wetland

Increasing success and operational efficiency of every technology is influenced by the intensity and situations of many variables and materials earlier, during and after activity of the technology. Reports have documented the influence of multiple component variables that have a major impact on success and effectiveness of the technology. In general, the variables like plant choice, choice of substrates, water depth, hydraulic loading rate (HLR), hydraulic retention time (HRT), feeding mood, electrode material, microorganisms and physiochemical parameters (moisture, temperature (T), dissolved oxygen (DO), oxidation-reduction potential (ORP), pH as well as chemical oxygen demand (COD), biochemical oxygen demand (BOD5), total suspended solids (TSS), total dissolved solids (TDS), densities of benzene, toluene, ethylbenzene including xylenes (BTEX) and so on) may be vital to the establishment of a successful CW program and effective treatment efficiency [255].

No universal assumption exists that one form of CW is superior than all the others. Nonetheless, there have been several research showings the efficiency of the various CW schemes with specific parameters of physicochemical characteristics like chemical oxygen demand (COD), biochemical oxygen demand (BOD), total suspended solids (TSS), benzene, toluene, ethylbenzene, including xylenes (BTEX) levels, and so on [256]. In Sternatia di Lecce, Italy, Gikas et al., [257] measured the cleansing efficacy of the BTEX wetland treatment cycle developed by SSF. Findings indicate that the amount of BTEX extraction varies from 46% to 55%. In addition, Ji et al., [258] managed heavy oil with an SSF scheme that provided water from China’s Liaohe Oilfield. Treatment efficacy was assessed, and the method showed higher mean extraction efficiency of 81%, 89%, and 89% separately for chemical oxygen demand (COD), biochemical oxygen demand (BOD) as well as mineral oil. Furthermore, Ji et al. [258] operated reed beds in FSF scheme-constructed wetlands for three years to manage heavy oil-producing water from China’s Liaohe Oilfield. Study demonstrated mean efficiencies in COD, BOD, and mineral oil extraction of 71, 77 including 92%, individually. Attempting to compare such two tests suggests that the SSF pattern was more effective in extracting COD and BOD than the FSF pattern, when the two flow schemes had almost the similar extraction efficacy for mineral oil [256].

In a phytotoxicity experiment to *Scirpus grossus*, two forms of flow scheme, free surface flow (FSF) along with sub-surface flow (SSF), were tested to choose a suitable way to extract total petroleum hydrocarbons (TPH) utilizing diesel as a hydrocarbon design. Comparison was made of the extraction efficiency of TPH for the two flow schemes. Mostly through simulation trials on many variables of wastewater were reported like temperature (T° C), dissolved oxygen (DO, mgL^-1^), oxidation-reduction potential (ORP, mV), as well as pH. In contrast, it also tracked average plant distances, wet weights as well as dry weights. Phytotoxicity analysis with plant *S. grossus* served at various diesel levels (1%, 2%, and 3%) for 72 days (V_diesel_/V_water_). An analysis of the dual flow schemes revealed that the SSF scheme was very effective in eliminating TPH from synthetic wastewater than the FSF scheme, with total extraction efficacy of 91.5 and 80.2 individually [256]. Also, for measurement of physical parameters, plant development as well as efficacy of total petroleum hydrocarbon (TPH) elimination as an indicator of diesel pollution, sampling was performed over 72 days of application to evaluate pilot-scale results. 4 pilot CWs with a horizontal subsurface flow scheme were implemented utilizing the *Scirpus grossus* bulrush. The CWs were filled with specific 0%, 0.1%, 0.2% along with 0.25% diesel levels (V_diesel_/V_water_). At the completion of 72 days, the TPH elimination efficiencies were 82%, 71% and 67% for 0.1%, 0.2%, and 0.25% diesel levels, individually. In comparison, the higher elimination efficacy of total suspended solids along with chemical oxygen demand (COD) are 100 as well as 75.4% separately, for 0.1% diesel. It has come to the conclusion that *S. grossus* is a promising plant that can be utilized to recover 0.1% diesel-polluted water in a very well-conducted CW [259].

This research comprised of an analysis with twelve wetland reactors running at various diesel levels of 0%, 0.1%, 0.175%, and 0.25% (V diesel/V water) and aeration levels (0, 1, and 2 L min^-1^) with the objective of assessing the impact of aeration delivery on the efficiency of pilot treatment over 72 days. The sub-surface flow constructed wetland (SSFCW) was cultivated with the indigenous *Scirpus grossus* plant in Malaysia. In the SSFCW reactors, the maximum extraction of total petroleum hydrocarbon (TPH) from diesel polluted water was identified to be 84.1%, 86.3%, and 88.3% for 0.1%, 0.175%, and 0.25%, individually, with 1 L min^-1^ aeration treatment. Aeration flow can also increase the development of plants and the bacterial community, suggesting that mixing plants and bacteria with aeration is a suitable solution for diesel polluted water. 1 L min^-1^ aeration is, as per statistical analyzes, a cost-effective operating parameter for removing TPH in diesel-polluted water utilizing *S. grossus* [260].

## 9. Optimization of Constructed Wetland

Optimization is conducted to try and define the optimal remedy for such situations and leads to procedure or engineered device performance enhancements and improvements [261]. Response surface methodology (RSM) has currently been implemented for optimization and modeling in several sectors like environmental analysis [262]. Response surface methodology (RSM) utilizes a mixture of multiple research design to produce mathematical models of various orders, for example linear, quadratic, cubic, respectively, to look for an optimal point from a unique collection of response variables including factors [263].

The RSM techniques were examined in this report, as well as statistical, appropriate and optimizing capabilities to boost As elimination by *L. octovalvis* uses a reed bed for operator. As elimination by *L. octovalvis* as a type of large-scale management, using RSM. Consequently, the major prospect of this examination is to find the ability of the RSM approaches to simulate and improve As elimination by *L. octovalvis* on a global scale.

### 9.1. Optimization Utilizing Response Surface Methodology (RSM)

A 2nd-order research design in response surface methodology, the Box–Behnken design was implemented to model the test to strengthen, establish and optimize the extraction of As from the soil [264]. The design was used to determine optimal situations for soil As density, sampling day including aeration rate for (percent) As extraction from soil by *L. octovalvis*. This research performed suitability tests for the model to decide if the remotely resembling model would yield incorrect performance [265]. 4 distinct large-degree polynomial models like linear, 2F1, quadratic as well as cubic techniques were constructed including adapted to the analytical findings to display the rapport among factors along with the response (total As extraction soil) [266]. In this analysis, multiple analyses were conducted to evaluate the appropriateness of models amongst different models, such as the consecutive model number of squares, model overview statistics including lack-of-fit tests. Comparison to other techniques, the quadratic model was important (*p* < 0.05), depending on the consecutive model number of squares. Therefore, the quadratic model was selected as the appropriate model. Constructed a quadratic model to match the obtained coefficients by multiple regression analysis. The quadratic model achieved is shown in the equation as follows:Total As removal from soil = + 67.22 − 4.82A + 5.28B − 1.22C − 3.75A^2^ + 10.25B^2^ − 17.86C^2^ − 1.36AB − 10.59AC − 2.73BC(1)

The extraction of As from soil varied from 34.25% to 87.07%. The overall soil As elimination (87.07%) was recorded in test 12 with the laboratory circumstances of soil As level (A, 5 ppm), day of sampling (B, 42 days) including aeration (C, 1 L min^-1^). The minimal soil As elimination (34.25%) was detected with the test soil density of As (A, 39 ppm), sampling day (B, 28 days) along with aeration rate (C, 2 L min^-1^). To obtain the highest performance in the extraction of As from the soil, the optimal system situation must be considered. The analysis of variance (ANOVA) shows that the measured F-value of model is 10.87964 and that the corresponding p value is < 0.05, demonstrating that the model is important [267]. There is only a probability of 0.24% that a “Model F-value” would happen due to noise [268]. The lack of fit was not significant as the p-value was 0.2564. The p-value was smaller than 0.05 for the model expected by Equation, indicating that it is significant for explaining the efficacy of As extraction from the soil. In this scenario, the independent factors in the soil (A) as well as sampling day (B) quadratic model of As level are very significant, as the p-value is smaller than 0.05. The model indicates an appropriate calculation coefficient (R^2^ = 0.930) as well as a modified calculation coefficient (modified-R [269] = 0.84) (Titah et al., [267]), suggesting that the model is sufficient to reflect the true correlation between the response and the relevant factors [270]. The model was utilized in pilot scale activity with optimum circumstances to calculate the quadratic impact and relationship of As elimination from the soil. The RSM model developed is suitable for estimating the As elimination output underneath the situations examined. The adequate estimation of the selected model was evaluated utilizing the experimental plots available in Design Expert 6.0.10 software, that involve the studentized residuals plotted versus normal probability, predicted versus studentized residuals, run versus studentized residuals together with run versus outlier [271]. Findings suggest that the studentized residuals plotted versus the normal probability displayed a straight line, indicating a normal distribution of the scientific findings and the predicted versus studentized residuals, run versus studentized residuals along with run versus outer all lie within the ±3.50 range, indicating that the model assumption was perfect beyond deviation [267].

As phytoremediation optimisation by *L. octovalvis* was performed in process of reed beds. Optimisation variables were loading rate (5, 22 and 39 ppm), retention time (14, 28, and 42 days), along with aeration rate flow (0, 1 and 2 L min^-1^). Assessment of results was focused on 6 findings; As in *L. octovalvis*, TF, translocation percentages, As extraction performance by *L. octovalvis*, As bioavailable, and total extractable As elimination in As-spiked sand. The optimization situation for As phytoremediation happened at As density of 39 ppm on Day 42 with 0.22 L min^-1^ founded on the response surface methodology utilizing Box–Behnken model. The optimal situation model allows the bioavailable as well as total extractable As at 94.8% and 72.6% separately. Some other responses had been As in *L. octovalvis* (1157.87 ppm), TF (1.62), translocation percentages (46%) and uptake achievement of As by *L. octovalvis*, (17.1%). The optimization condition error for all responses was less than 10% by comparison the model with a validation run. Variables like loading, retention time, and aeration flow may influence the output of As phytoremediation by *L. octovalvis* [272].

Phytoremediation utilizing the wild plant *Melastoma malabathricum* L was examined as a compared to natural approaches of treatment for remediating the accumulation of lead (Pb) in soil. In the current analysis, ingestion of lead (Pb) by *M. malabathricum* L. was assessed along with standardized utilizing Response surface methodology (RSM). The Box–Behnken design (BBD) was utilized to optimize lead extraction of Pb, with 3 primary factors utilized in the optimization (Pb intensity in sand: 20–70 ppm; exposure time: 14–70 days; aeration rate: 0–3 L min^-1^). For the Pb intensity in sand, the expected optimal parameters were 44.1 ppm Pb, an intake time of 14 days as well as 0 L min^-1^ aeration rate, with a real Pb bioaccumulation of 3596.0 ppm. This result correlates comfortably with the expected RSM value (3855.1 ppm Pb). The discrepancy within the validation value including the expected value was around 6.7%, suggesting that RSM could accurately calculate with very small deviation the optimal Pb bioaccumulation. The bioaccumulation of maximum Pb can be accomplished without aeration necessity, culminating in a cost-effective management method [273].

The optimization of the elimination of COD from the sewage of palm oil mill (POME) utilizing the Reverse Osmosis (RO) membrane was examined. The Box Behken design was efficiently used to obtain observational situations for reducing POME’s COD value. Based on the polynomial regression model, the values of an acting variables (POME density, pH, and transmembrane stress) were adjusted. POME intensity (vol. %) = 28.30, pH = 10.75 and transmembrane pressure = 0.69 kPa were observed to be the expected situations for generating smaller COD values. The COD value predicted was 24.137 ppm that was achieved in better agreement with experimental value as 25.763 ppm [274].

This research examined optimal situations for the complete extraction of petroleum hydrocarbons from diesel-polluted water utilizing *Scirpus grossus* with phytoremediation technique. Also, the extraction of TPH from sand was introduced as a secondary response. The optimal parameters for highest extraction of TPH were calculated via a Box–Behnken design. Three operating factors, i.e., the intensity of diesel (0.1%, 0.175%, 0.25% V_diesel_/V_water_), the aeration level (0, 1 as well as 2 L min^-1^), along with the retention time (14, 43, as well as 72 days), were observed by establishing the highest TPH extraction as well as diesel intensity, the retention period within the specified limit and the minimal aeration level. The optimal situations were observed to be a 0.25% diesel intensity (V_diesel_/V_water_), a 63-day retention period and no aeration with an average 76.3% and 56.5% gross extraction of TPH from water and sand, accordingly. From a confirmation study of the optimal situations, it was concluded that the highest extraction of TPH from polluted water and sand was 72.5% and 59% separately, that was a variation of 5% along with 4.4% from the values provided by the Box–Behnken method, showing that *S. grossus* is an indigenous Malaysian plant that is capable of remediating hydrocarbon-containing wastewater [270].

This research explored the efficacy of a biologically aerated filter (BAF) as an external treatment for concurrent chemical oxygen demand (COD), ammonium (NH_4_^+^-N), and manganese (Mn^2+^) extraction in drinking water management plant schemes. The experimental design was a face-centered central composite design (FC-CCD) with three major factors: COD load, aeration rate (AR) including HRT. Optimized situations for maximal extraction of COD, NH_4_^+^-N including Mn^2+^ were calculated via the methodology of response surface that COD load was fixed as the limit whereas aeration rate along with hydraulic retention time were decreased. The optimal situations were observed to be COD load of 0.90 kg m^-3^, AR of 0.30 L min^-1^ along with HRT of 7.47 h with subsequent elimination of COD, NH_4_^+^-N, including Mn^2+^ as 95.5%, 93.9%, along with 94.8%, individually. These optimal scenarios were utilized to calculate BAF process investment and operational costs for a capability of 100,000 m^3^ day^-1^. The average gross investment and operational expenses were US$ 8,110,600 and US$ 0,022 per m^3^, collectively [271].

In this experiment the extraction of As by *Ludwigia octovalvis*, was optimized in a pilot reed bed. A Box–Behnken design was utilized for the forecasting of highest As extraction, along with a comparison study of both Response Surface Methodology (RSM) including an artificial neural network (ANN). The predicted optimal situation utilizing all models’ marketability feature was 39 ppm for soil As density, an average period of 42 days (the sampling day) and an aeration rate of 0.22 L min^-1^, with the predicted As extraction values of 72.6% and 71.4% simultaneously by RSM and ANN. The testing of the maximum expected point demonstrated a real As extraction of 70.6%. It was done with the variance of 3.49% (RSM) and 1.87% (ANN) between the validity value and the expected values. The RSM and ANN model efficiency comparison demonstrated that ANN results smarter than RSM with greater R2 (0.97) values near 1.0 and very low Average Absolute Deviation (AAD) (0.02) and Root Mean Square Error (RMSE) values near 0. All models were suitable for optimizing As elimination with ANN showing substantially greater reliable and suitable capability than RSM [267].

This study investigated the effectiveness of a biological aerated filter (BAF) as an additional treatment in drinking water treatment plant systems for simultaneous chemical oxygen demand (COD), ammonium (NH_4_^+^–N) and manganese (Mn^2+^) removal. The experimental design was face centered-central composite design (FC-CCD) with three operational variables: COD load, aeration rate (AR) and hydraulic retention time (HRT). Optimum conditions for maximum COD, NH_4_^+^–N and Mn^2+^ removal was determined through response surface methodology, where COD load was set as the maximum while aeration rate and hydraulic retention time were minimized. The optimum conditions were found to be COD load of 0.90 kg/m^3^, AR of 0.30 L/min and HRT of 7.47 h with predicted simultaneous COD, NH_4_^+^–N, and Mn^2+^ removal as 95.5%, 93.9%, and 94.8%, respectively [271].

This study investigated the optimum conditions for total petroleum hydrocarbon (TPH) removal from diesel-contaminated water using phytoremediation treatment with *Scirpus grossus*. In addition, TPH removal from sand was adopted as a second response. The optimum conditions for maximum TPH removal were determined through a Box–Behnken design. Three operational variables, i.e. diesel concentration (0.1%, 0.175%, 0.25% V_diesel_/V_water_), aeration rate (0, 1, and 2 L/min) and retention time (14, 43, and 72 days), were investigated by setting TPH removal and diesel concentration as the maximum, retention time within the given range, and aeration rate as the minimum. The optimum conditions were found to be a diesel concentration of 0.25% (V_diesel_/V_water_), a retention time of 63 days and no aeration with an estimated maximum TPH removal from water and sand of 76.3% and 56.5%, respectively. From a validation test of the optimum conditions, it was found that the maximum TPH removal from contaminated water and sand was 72.5% and 59%, respectively, which was 5% and 4.4% deviation from the values given by the Box–Behnken design [270].

### 9.2. Determination of the Optimal Point Utilizing the Desirability Function Method

The optimal situation imagined through RSM has been calculated using the desirability function process [275]. This approach involves expectations as well as preferences for every one of the factors along with provides a protocol for deciding the correlation for each factor between expected As elimination and the desirability of the responses. The factors were adjusted as below while performing on the pilot scale: highest soil As level, highest sampling day and minimal aeration rate [270]. By utilizing numerical optimization operations in the Design Expert software 6.01, 4 explanations with a desirability value higher than 0.860 were recommended for the optimum situation. It indicates a desirability of 0.922 for highest extraction of As from soil efficiencies. The overall As extraction of 72.6% was gained in soil (A), sampling day 42 (B) including 0.22 L min^-1^ for aeration rate (C) under optimized situations of 39 mg kg^-1^ As. During optimal scenarios the RSM projected a cumulative As extraction happening at 71.4%. The finding of the validation demonstrated that As extraction for RSM was gained at nearly 70.06%. The validation values were observed to be in excellent correlation with the predicted values, implying the satisfaction of the model acquired to optimize the extraction of the As. The difference between the validity value including the expected values was within 3.49% including 1.87% separately. So, it can be inferred that optimizing RSM model pilot-scale As extraction was sufficient [267].

This study investigated the optimum conditions for total petroleum hydrocarbon (TPH) removal from diesel-contaminated water using phytoremediation treatment with *Scirpus grossus*. The desirability function methodology was used for this optimization. Three operational variables, i.e., diesel concentration (0.1%, 0.175%, 0.25% V_diesel_/V_water_), aeration rate (0, 1 and 2 L/min) and retention time (14, 43, and 72 days), were investigated by setting TPH removal and diesel concentration as the maximum, retention time within the given range, and aeration rate as the minimum. By using the function of numerical optimization in the Design Expert software, we found a desirability of 0.883 for the maximum TPH removal efficiencies. The maximum TPH removal from water and sand was 76.3% and 56.5%, respectively, at the optimized conditions of 0.25% diesel concentration, 63 days retention time and 0 L/min aeration rate. Validation experiments were carried under at the optimum conditions to confirm the predicted optimum response. The measured results under the optimum conditions were about 72.5% for TPH removal from water and 59% for TPH removal from sand. These results were very close to the predicted values, indicating the adequacy of the obtained model to optimize TPH removal from water and sand. The deviations between the measured and predicted values were within 5%. Therefore, it can be concluded that the regression models were appropriate in their reduced forms [270].

### 9.3. Three-Dimensional (3D) Surface Plot under the Optimum Conditions

The response surfaces of the maximum extraction of As from soil by *L. octovalvis* was developed utilizing a quadratic model as a mathematical formula. The correlations between As density in soil (A), day of sampling (B) and rate of aeration (C). The 3D surface plot of the impact of As density in soil (A) including sampling day (B) on the extraction of As. The As level in the 5–39 ppm range can be found to took arsenic elimination to a steady state. This may probably be due to the variety identified involving levels of As that were not substantially dissimilar. The impact of sampling day (B) was a feedback rise within 14–42 days only [267]. The older sampling day is assumed to result in greater elimination of As [276]. After the aeration rate was risen, it was found that As extraction from soil was expanded. It has been shown that the As elimination from the soil rose whenever the aeration rate raised from 0 to 1.5 L min^-1^ and subsequently gradually declined after the aeration rate raised to 2 L min^-1^; the aeration rate remained efficient for the elimination of As from the soil, however with the level raised to 2 L min^-1^ [267]. The exact extraction of As from soil was 60% at an aeration rate of 1.5 L min^-1^, that is adequate in the pilot method to raise the accessibility of oxygen in soil to increase microbiological activity across the root [277]. The phytoremediation output by *L. octovalvis* that not only rely on the plant themselves but also on the relationship between the plant roots and the microorganisms. The influence of sampling day (B) including aeration rate (C) on the extraction of As from soil, it was found that As elimination was improved with an improvement in aeration rate, but the sampling day impact was not substantially dissimilar (*p* > 0.05). This illustrates that aeration rate is critical for improving the extraction of As from the soil. A later day of sampling including adequate aeration rate are quite necessary for phytoremediation to enhance the extraction of contaminants. Since we wanted a very efficient treatment device, the minimal aeration rate had been established [270].

In this study, the aim of optimization was to find the conditions that provide maximum TPH removal from water in a phytoremediation process. The three-dimensional (3D) response surface is a graphical representation of the regression function. The 3D response surface plots show how TPH removal from water and sand relate to the factors of diesel concentration, retention time and aeration rate through quadratic model equations. It was observed that TPH removal from water and sand increased with an increase in the aeration rate. It was clearly shown that there was a steady increase in the TPH removal from water when the aeration rate was increased from 0 to 1 L/min and later gradually decreased when the aeration was further increased to 2 L/min. The aeration rate was effectively positive until 2 L/min for TPH removal from sand. The actual TPH removal from water was 88.3% with an aeration rate of 1 L/min, which is enough on the pilot scale with a subsurface flow system to enhance oxygen availability for biodegradation. However, the maximum actual TPH removal from sand (67.3%) was achieved with an aeration rate of 2 L/min. TPH removal increased with an increase in the aeration rate, but the effect of aeration rate was not significantly different (*p* > 0.05) when applying a high level of 2 L/min. This demonstrates that longer retention time reduces more TPH concentration in water, but the diesel concentration and aeration rate do not have much effect on TPH removal efficiency [270].

## 10. Direction for Future Research, Challenges and Overcomes

It has been broadly acknowledged that after decades of research as well as application, CWs are an effective management system for different wastewaters. The present analysis demonstrates that improvements in the design as well as operation of CWs, have significantly enhanced the efficiencies in the elimination of pollutants, and have enhanced the renewable implementation of this management process. For instance, a reasonable modification of the hydraulic design, mode of operation, contaminant loading rate, along with likely choice of plants including substrates may attain the outstanding efficiency in CWs for curing max intensity of wastewater or during winter.

### 10.1. Direction for Future Research

Direction for future research on the design along with operation of CWs for treating wastewater are provided in Table 6. 

### 10.2. Challenges and Overcome

Provided the progressively stringent water safety requirements for wastewater management and water recycle globally, moreover, CWs also has few challenges, as well as more study and innovation work are needed. To sum it up (Figure 3):

Choice of plants and substrates shows that wetland phytoplankton including substrates are already key to the effective elimination of pollutants from wastewater in CW. Careful choice of macrophyte varieties should be given much emphasis (such as broad formation of vegetation, abundant) delivery of O_2_ and C substances, greater absorption of toxins, particularly evolving toxins, like heavy metals and medical devices, strong contaminant overload resistance) implemented in CWs in warm and cool environments for the management of wastewater, complex and expensive assessment of varieties and seasonal variations is further required. In conjunction, certain non—traditional wetland materials should be produced and utilized for CWs (manufacturing by product, farm garbage, and so on) that has excellent adsorption potential and is advantageous for removing techniques.The analysis of design and operational parameters reveals that the optimum efficiency of the treatment depends fundamentally on ecological, hydraulic, and operational situations. Consequently, in future research, improving these situations requires comprehensive and detailed examination. In addition, attention should also be given to researching the main pathway and technique relating to stronger contaminant elimination.Considering the ongoing progress of study and functional implementation in conventional CWs, in future experiments, innovative techniques and approaches for improving wastewater implemented in CWs are urgently needed to augment the effective water freshness. Such techniques and approaches usually includes: artificial ventilation, tidal activity, phase feeding, extrinsic incorporation of C, microorganisms increase, distribution of different plants, mixture of different substrates, CWs, and hybrid CWs, respectively.Vitamins and minerals as well as other contaminants integrated by wetland plants are recorded to be able to discharge into water whenever plants die and decompose however during snowy weather, that may contribute to the low elimination efficiency in CWs. Experiment and modernization on effective techniques for plant harvesting, and purification and regeneration of plant materials in CWs are therefore important.

## 11. Conclusions

This review-based study illustrates that the factors for CW design and operation such as plant selection, substrate selection, water depth, loading rate, hydraulic retention time, and feeding mode are crucial to achieve the sustainable treatment performance. Wetland macrophytes and substrates represent two factors that influence the efficiency of pollutant removal in CWs. More attention should be given to appropriate plant species selection for CWs. An intensive evaluation of differences between species and season is also needed. In addition, some non-conventional wetland media, characterized by high sorption capacity, should be studied and used for CWs. Moreover, the review of the design and operating parameters shows that the optimal treatment performance is vitally dependent on environmental, hydraulic and operating conditions. Therefore, understanding how to manage and optimize these conditions warrants more investigation. Additional research on the critical pathways and mechanisms corresponding to higher pollutant removal should be taken into consideration. The review of design and operation parameters (plant and substrate selection, and hydraulic conditions) shows that the optimal treatment performance is crucially dependent on hydraulic, environmental, and operating conditions. Therefore, if an optimization of the design and management of these systems wants to be accomplished, further studies on the aspects above mentioned would be needed. For the sake of clarification, it is worth mentioning that as well as studies on design and operation parameters, additional research on maintenance processes and new strategies and technologies is necessary for sustainable CW systems and water quality improvement. Taking into account the efficient and sustainable implementation of full-scale CWs, future studies should focus on a comprehensive assessment of plants and substrates in field trials under real conditions, optimization of environmental and operational parameters, exploration of novel enhancement technologies and maintenance strategies. New studies will provide information that will increase the successful application and sustainability of CWs.

## Figures and Tables

**Figure 1 ijerph-17-08339-f001:**
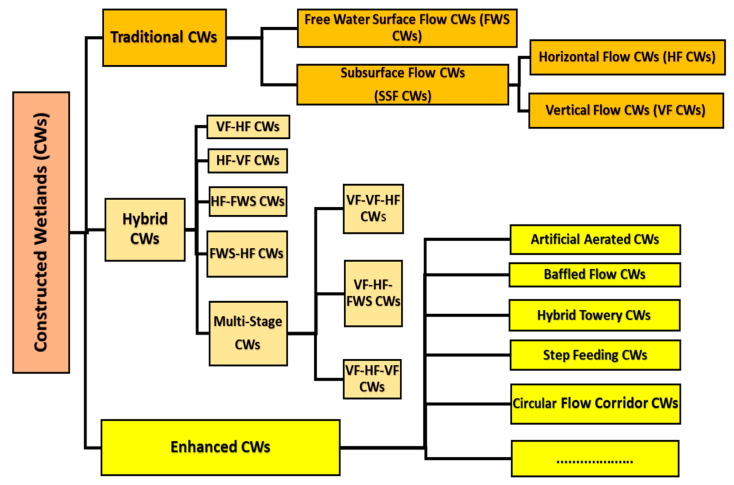
Classification of CWs utilized in wastewater management.

**Figure 2 ijerph-17-08339-f002:**
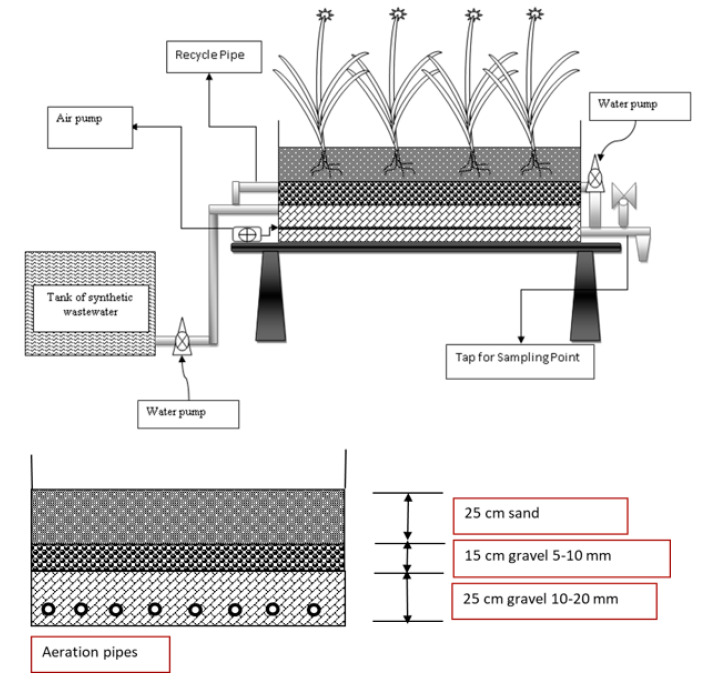
Design of Constructed Wetland.

**Figure 3 ijerph-17-08339-f003:**
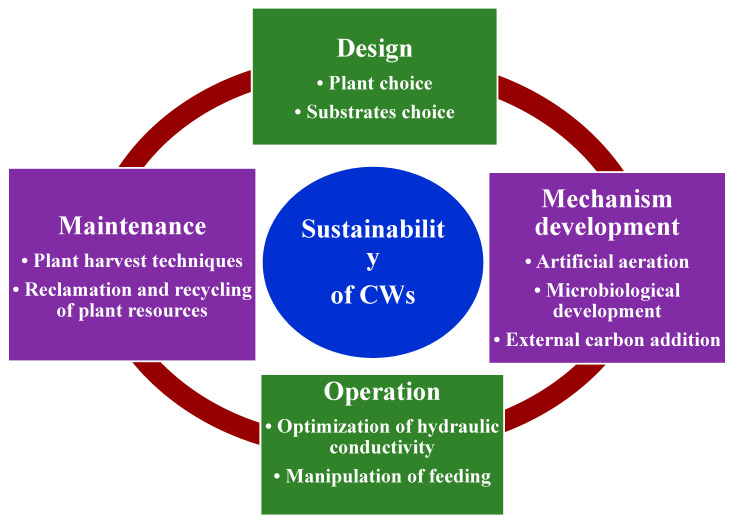
Summary of Assumptions for achieving sustainable development of CWs.

**Table 1 ijerph-17-08339-t001:** The design, operational, and physicochemical parameters of studied constructed wetlands (CWs) and corresponding references.

Design, Operational, and Physicochemical Parameters	References
Operational Factors	
Hydraulic loading rate	[54]
Organic loading rate	[55]
Hydraulic retention time	[56]
Physicochemical Parameters	
pH	[57]
Temperature	[58]
Dissolved oxygen	[58]
Planted and Unplanted CWs	[59]
Role of Support Matrix	[60]
Effect of Seasonality (summer and winter)	[61]

**Table 2 ijerph-17-08339-t002:** List of developing countries (According to World Bank, [69]).

Upper middle income (between $4086 and $12,615 GNI per capita)	Lower middle income (between $1036 and $4085 GNI per capita)	Low-income (less than $1035 GNI per capita)
Albania, Algeria, Angola, Argentina, Azerbaijan, Belarus, Bosnia and Herzegovina, Botswana, Brazil, Bulgaria, China, Colombia, Costa Rica, Cuba, Dominican Republic, Ecuador, Gabon, Hungary, Iran Islamic Republic, Iraq, Jamaica, Jordan, Kazakhstan, Lebanon, Libya, Malaysia, Mauritius, Mexico, Montenegro, Namibia, Panama, Peru, Romania, Serbia, South Africa, Thailand, The former Yugoslav Republc of Macedonia, Tunisia, Turkey, Turkmenistan, Venezuela, RB.	Armenia, Bolivia, Cameroon, Cape Verde, Congo, Côte d’Ivoire, Djibouti, Egypt, El Salvador, Georgia, Ghana, Guatemala, Guyana, Honduras, India, Indonesia, Lesotho, Mauritania, Moldova, Morocco, Nicaragua, Nigeria, Pakistan, Papua New Guinea, Paraguay, Philippines, São Tomé and Principe Senegal, Sri Lanka, Sudan, Syrian Arab Republic, Ukraine, Uzbekistan, Vietnam, Yemen Rep., Zambia.	Bangladesh, Benin, Burkina Faso, Burundi, Central African Republic, Chad, Comoros, Democratic Republic of the Congo, Eritrea, Ethiopia, Gambia, The Guinea Guinea-Bissau, Haiti, Kenya, Kyrgyz Republic, Liberia, Madagascar, Malawi, Mali, Mozambique, Myanmar, Nepal, Niger, Rwanda, Sierra Leone, Somalia, Tajikistan, Tanzania, Togo, Uganda, Zimbabwe.

**Table 3 ijerph-17-08339-t003:** Ornamental flowering plants and removal of wastewater pollutants in CWs (constructed wetlands) around the globe.

Country	Type of Wastewater	Vegetation	Removal Efficiency of Pollutants (%)	Reference
Brazil	Domestic	*Heliconia psittacorum*	TSS: 88, COD: 95, BOD: 95	[96]
	Domestic	*Alpinia purpurata, Arundina bambusifolia, Canna* spp., *Heliconia psittacorum* L.F.	COD: 48–90, PO_4_-P: 20, TKN: 31 and TSS: 34.	[97]
	Swine	*Hedychium coronarium, Heliconia rostrata*	COD: 59, TP: 44, TKN: 34 and NHx: 35 COD: 57, TP: 38, TKN: 34 and NHx: 37	[98]
		*Hemerocallis flava*	COD: 72, BOD: 90, TN: 52, TP: 41 and SST: 72.	[99]
		*Heliconia psittacorum* L.F		[100]
China	Municipal	*Canna indica*	COD: 77, BOD: 86, TP: > 82, TN: > 45	[101]
	Aquaculture ponds	*Canna indica* mixed with other species	BOD: 71, TSS: 82, chlorophyll-a: 91.9, NH_4_-N: 62, NO_3_-N: 68 and TP: 20.	[102]
	Domestic	*Canna indica* Linn	COD: 82.31, BOD: 88.6, TP: > 80, TN: > 85	[103]
	Municipal	*Canna indica*	NH_4_-N: 99, PO_4_-P: 87	[104]
	Drain of some factories	*R. carnea, I. pseudacorus, L. salicaria*	COD: 58-92, BOD: 60–90 TN: 60–92, TP: 50–97,	[105]
	River	*Canna* sp.	COD: 95, N-NH_4_: 100, N-NO_3_: 76, TN: 72	[106]
	Domestic	*Canna indica*	TP: 60, NH_4_-N: 30–70, TN:~25	[34]
	Aquaculture ponds	*Canna indica* mixed with other natural wetland plants	BOD: 56, COD: 26, TSS: 58, TP: 17, TN: 48 and NH_4_-N: 34.	[53]
	Wastewater from a student dormitory (University)	*Canna indica* mixed with other natural wetland plants	COD: 50–70, BOD: 60–80, N-NO_3_: 65–75, TP: 50–80	[107]
**Country**	**Type of Wastewater**	**Vegetation**	**Removal Efficiency of Pollutants (%)**	**Reference**
China	Domestic	*Canna indica* and *Hedychium coronarium*	TP: 40–70	[108]
	Polluted river	*Iris pseudacorus* mixed with other natural wetland plants	TN: 68, NH_4_-N: 93, TP: 67	[109]
	Sewage	*Iris pseudacorus,* mixed with other plants of natural wetlands	TN: 20 and TP: 44	[110]
	Municipal	*Canna indica*	COD: 60, NO_3_-N: 80, TN: 15, TP: 52	[111]
	Simulated polluted river water	*Iris sibirica*	COD: 22, TN: 46, NH_4_-N: 62, TP: 58	[112]
	Synthetic	*Canna* sp.	Fluoride: 51, Arsenic: 95	[26]
	Simulated polluted river water	*Iris sibirica*	Cd: 92	[113]
	Synthetic	*Canna indica* L.	N: 56–60	[114]
	Synthetic (hydrophonic sol.)	*Canna indica* L.	TN: 40–60, N-NO_3_: 20–95, NH_4_-N: 20–55	[115]
Chile	Sewage	*Zantedeschia aethiopica*, *Canna* spp. and *Iris* spp.	BOD: 82, TN: 53, TP: 60.	[116]
	Sewage	*Tulbaghia violácea,* and *Iris pseudacorus.*	BOD: 57–88, COD: 45–72, TSS: 70–93, PO_4_ -P: 6–20.	[117]
	Ww rural community	*Zantedeschia aethiopica*	Organic matter: 60%, TSS: 90%	[118]
Colombia	Domestic	*Heliconia psíttacorum*	NH_3_: 57 COD: 70	[119]
	Synthetic landfill leachate	*Heliconia psittacorum*	COD, TKN and NH4 (all: 65–75)	[120]
	Cattle bath	*Alpinia purpurata*	SST: 58, TP: 85, COD: 63	[121]
	Municipal	*Heliconia psitacorum*	Bisphenol A: 73, Nonylphenols: 63	[122]
Costa Rica	Dairy raw manure	*Ludwigia inucta, Zantedechia aetiopica, Hedychium coronarium* and *Canna generalis*	BOD: 62, NO_3_ -N: 93, PO_4_-P: 91, TSS: 84	[123]
Egypt	Municipal	*Canna* sp.	TSS: 92, COD: 88, BOD: 90	[124]
	Municipal	*Canna* sp.	TSS: 92, COD: 92, BOD: 92	[125]
**Country**	**Type of Wastewater**	**Vegetation**	**Removal Efficiency of Pollutants (%)**	**Reference**
India	Paper mill effluent	*Canna indica*	9,10,12,13-tetrachlor-ostearic acid: 92 and 9,10-dichlorostearic acid: 96	[126]
	Synthetic	*Canna indica*	Dye: 70–90 COD: 75	[127]
	Synthetic greywater	*Heliconia angusta*	COD:40, BOD: 70, TSS: 62, TDS: 19	[128]
	Domestic	*Canna generalis*	TN: 52, T-PO_3_: 9	[129]
	Collection pond	*Canna Lily*	BOD: 70–96, COD: 64–99	[130]
	Hostel greywater	*Canna indica*	COD, TKN and Pathogen all up 70	[131]
	Domestic	*Polianthus tuberosa* L.	Heavy metals (Pb and Fe: 73–87), (Cu and Zn: 31–34) and Ni and Al: 20–26	[132]
Ireland	Domestic	*Iris pseudacorus*	TN: 30, TP:28	[133]
Italy	Synthetic	*Zantedeschia aethiopica, Canna indica*	N: 65–67, P: 63–74, Zn and Cu: 98–99, Carbamazepine: 25–51, LAS: 60–72	[134]
Kenya	Flower farm	*Canna* spp.	BOD: 87, COD: 67, TSS: 90, TN: 61	[135]
Mexico	Municipal	*Zantedeschia aethiopoca*	COD: 35, TN: 45.6	[136]
	Domestic	*Zantedeschia Aethiopica* and *Canna flaccid*	SST: 85.9, COD: 85.8, NO_3_ -N: 81.7, NH_4_-N: 65.5, NT: 72.6	[137]
	Coffee processing	*Heliconia psittacorum*	COD: 91, Coliformes: 93	[138]
	Domestic	*Strelitzia reginae, Zantedeschia esthiopica, Canna hybrids, Anthurium andreanum, Hemerocallis Dumortieri*	COD: > 75, P: > 66, Coliforms: 99	[139]
	Domestic	*Zantedeschia aethiopica*	BOD: 79, TN: 55, PT: 50	[140]
	Wastewater form canals	*Zantedeschia aethiopica*	COD: 92, N-NH_4_: 85, P-PO_4_: 80	[141]
	Municipal	*Strelitzia reginae, Anthurium, andreanum.*	TSS: 62, COD: 80, BOD: 82, TP: > 50, TN: > 49	[142]
	Groundwater	*Zantedeschia aethiopica* and *Anemopsis californica*	As: 75–78	[142]
	Domestic	*Gladiolus* spp.	BOD: 33, TN: 53, TP: 75	[143]
**Country**	**Type of Wastewater**	**Vegetation**	**Removal Efficiency of Pollutants (%)**	**Reference**
Mexico	Mixture of greywater (from a cafeteria and research laboratories)	*Zantedeschia aethiopica* and *Canna indica*	COD: 65, NT: 22.4, PT: 5.	[144]
	Domestic	*Zantedeschia aethiopica*	BOD: 70	[145]
	Domestic	*Heliconia stricta, Heliconia psittacorum* and *Alpinia purpurata*	BOD: 48, COD: 64, TP: 39, TN: 39	[146]
	Municipal	*Canna hybrids* and *Strelitzia reginae*	DQO: 86, NT: 30-33, PT: 24–44	[147]
	Municipal	*Zantedeschia aethiopica* and *Strelitzia reginae*	COD: 75, TN: 18, TP: 2, TSS: 88.	[148]
	Domiciliar	*Spathiphyllum wallisii, Zantedechia aethiopica, Iris japonica, Hedychium coronarium, Alocasia* sp., *Heliconia* sp. and *Strelitzia reginae*	N-NH_4_: 64–93 BOD: 22–96 COD: 25–64	[149]
	Community	*Zantedeschia aethiopica, Lilium* sp., *Anturium* spp. and *Hedychium coronarium*	NT: 47, PT: 33, COD: 67	[150]
	Stillage Treatment	*Canna indica*	BOD: 87, COD: 70	[151]
	Artificial	*Iris sibirica* and *Zantedeschia aethiopica*	Carbamazepine: 50–65	[152]
	Community	*Alpinia purpurata* and *Zantedeschia aethiopica*	---	[153]
	Polluted river	*Zantedeschia aethiopica*	NO_3_-N: 45, NH_4_-N: 70, PO_4_-P: 30	[154]
	Municipal	*Spathiphyllum wallisii,* and *Zantedeschia aethiopica*	---	[155]
	University	*Strelitzia reginae*	---	[156]
Nepal	Municipal	*Canna latifolia*	TSS: 97, COD: 97, BOD: 89, TP: > 30	[157]
Portugal	Tannery	*Canna indica* mixed with other plants	COD: 41–73, BOD: 41–58	[158]
	Community	*Canna flaccida, Zantedeschia aethiopica, Canna indica, Agapanthus africanus* and *Watsonia borbonica*	BOD, COD, P-PO_4_, NH_4_ and total coliform bacteria (all up to 84)	[159]
**Country**	**Type of Wastewater**	**Vegetation**	**Removal Efficiency of Pollutants (%)**	**Reference**
Spain	Domestic	*Iris* spp.	Bacteria: 37	[160]
	Municipal	*Iris pseudacorus*	Bacteria: 43	[161]
Sri Lanka	Municipal	*Canna iridiflora*	BOD: 66, TP: 89, NH_4_-N: 82, N-NO_3_: 50	[162]
Taiwan	Domestic	*Canna indica*	N-NH_4_: 73, BOD: 11	[163]
		*Canna indica*	N-NH_4_: 57, N-NO_3_: 57	[164]
Thailand	Domestic	*Canna* spp.	COD: 92, BOD: 93, TSS: 84, NH_4_-N: 88, TP: 90	[165]
	Seafood	*Canna siamensis, Heliconia* spp. and *Hymenocallis littoralis*	BOD: 91–99, SS: 52–90, TN: 72–92 and TP: 72–77	[166]
	Domestic	*Heliconia psittacorum* L.f. and *Canna generalis* L. Bailey	TSS: Both > 88, COD: 42–83	[167]
	Fermented fish production	*Canna hybrid*	BOD, COD, TKN: ~ 97	[168]
	Collection system for business and hotel	*Cannae lilies, Heliconia*	BOD: 92, TSS: 90, NO_3_-N: 50, TP: 46	[169]
	Domestic	*Crinum asiaticum, Spathiphyllum clevelandii* Schott	PO4-P: ~20	[170]
Turkey	Municipal	*Iris australis*	NH_4_-N: 91, NO_3_-N: 89, TN: 91	[171]
USA	Domestic	*Canna flaccida, Gladiolus* sp., *Iris* sp.	Baceria: ~50	[172]
	Nursery	*Canna· generalis, Eleocharis dulcis, Iris Peltandravirginica.*	N: ~50, P: ~60	[173]
	Domestic	*Iris pseudacorus* L., *Canna x. generalis* L.H. Bail., *Hemerocallis fulva* L. and *Hibiscus moscheutos* L	BOD > 75, TSS > 88, Fecal baceteria > 93	[174]
	Tilapia production	*Canna* sp.	TSS: 90, NO_2_-N: 91, NO_3_-N: 76, COD: 12.5 and NH_3_-N: 7.5	[175]
	Stormwater runoff	*Canna x generalis* Bailey, *Iris pseudacorus* L., *Zantedeschia aethiopica* (L.)	N and P Canna (>90), Iris (>30) Zantedeschia (>90)	[176]
**Country**	**Type of Wastewater**	**Vegetation**	**Removal Efficiency of Pollutants (%)**	**Reference**
	Residential	*Aeonium purpureum* and *Crassula ovate, Equisetum hyemale, Nasturtium, Narcissus impatiens,* and *Anigozanthos*	TSS: 95 BOD: 97	[177]
Vietnam	Fishpond	*Canna generalis*	BOD: 50, COD: 25–55	[74]
United Kingdom	Herbicide polluted water	*Iris pseudacorus*	Atrazine: 90–100	[178]

TSS= Total Suspended Solids; COD= Chemical Oxygen Demand; BOD= Biological Oxygen Demand; PO_4_-P= Orthophosphate as Phosphorus; TKN= Total Kjeldahl Nitrogen; TP= Total Phosphorus; NHx= Na^+^/H^+^ Antiporters; TN= Total Nitrogen; SST= Sho-Saiko-to (SST) Oriental Medicine; NH_4_-N= Ammonium-Nitrogen; NO_3_-N= Nitrate-Nitrogen; N-NH_4_= Ammonium-Nitrogen; N-NO_3_= Nitrate-Nitrogen; T-PO_3_= Total Phosphite; DQO= Demanda Química de Oxígeno (Spanish: Chemical Oxygen Demand).

**Table 4 ijerph-17-08339-t004:** Four most commonly genera plants used in CWs around the globe, identified during the 87 survey studies in 21 countries, grouped by continents.

			America				
	Asia	Europe	North America		Central and South America	Africa	Total
			USA	Mexico			
Canna	22	4	5	4	2	2	39
Iris	5	5	4	2	2		18
Heliconia	4			4	4		12
Zantedeschia		2	1	13	3	1	20

**Table 5 ijerph-17-08339-t005:** Substrates typically chosen for management of wastewater in CW.

Kinds of Substrates	Source
Natural Material	
Sand	[202]
Gravel	[203]
Clay	[203]
Calcite	[204]
Marble	[38]
Vermiculite	[38]
Bentonite	[205]
Dolomite	[204]
Limestone	[206]
Shell	[207]
Shale	[208]
Peat	[208]
Wollastonite	[209]
Maerl	[208]
Zeolite	[210]
**Industrial by-product**	
Slag	[34]
Fly ash	[205]
Coal cinder	[211]
Alum sludge	[212]
Hollow brick crumbs	[211]
Moleanos limestone	[213]
Wollastonite tailings	[214]
Oil palm shell	[199]
**Artificial products**	
Activated carbon	[211]
Light weight aggregates	[208]
Compost	[208]
Calcium silicate hydrate	[201]
Ceramsite	[201]
**Others**	
Alum sludge	[215]
Apatite material	[216]
Biochar	[217]
Bauxite	[218]
Construction wastes	[219]
Tire chips	[220]
Polyethylene terephthalate (PET)	[221]
Filtralite	[222]
Oyster shell	[223]
PHBV and PLA blend Wood mulch	[224] [225]
Rice straw	[226]


**Table 6 ijerph-17-08339-t006:** Direction for future research on design and operation of CWs.

Parameter	Design Criteria	
	FWS CWs	SSF CWs
Bed size (m^2^)	Larger if available	<2500
Length to width ratio	3:1–5:1	<3:1
Water depth (m)	0.3–0.5	0.4–1.6
Hydraulic slope (%)	<0.5	0.5–1
Hydraulic loading rate (m/day)	<0.1	<0.5
Hydraulic retention time (day)	5–30	2–5
Media	Natural media and industrial by-product preferred, porosity 0.3–0.5, particle size <20 mm (50–200 mm for the inflow and outflow)	
Vegetation	Native species preferred, plant density 80% coverage	

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
