# Peer review of "Design, Operation and Optimization of Constructed Wetland for Removal of Pollutant"

_ijerph, 2020, doi:10.3390/ijerph17228339_

Round 1
Reviewer 1 Report
The author took large parts to describe the constructed wetland, like desgn, operation and optimization, but lack of efficient evidence for the utilization of the constructed wetland. Thus, I suggested that the author could describe some more about the application of constructed wetland in the treatment of pollutant removal.
Reviewer 2 Report
The manuscript is focused in the review of constructed wetlands for the removal of different pollutants.In general the information need to be ordered and presented in a more clear way. Tables with the main information reviewed from literature would help to improve the manuscript. Organizing the information according to the type of wastewater, type of constructed wetland, operational parameters, vegetation, etc. Some sections are focused only in information of metal removal, and information regarding other types of pollutants removal in wastewater should be included, otherwise authors should delimit the review (organize the information). Authors please check abbreviations, define in first mention and then use it, also check the use of italics and the correct use of "which" word.
Some other comments: > Abstract: Should include the contribution of the review for example, perspectives and challenges and major advances in the field. - Page 1 Line 38 check expression for grammar -Page 2 Line 73. Chech "poison" word meaning and use, do you refer to toxic compound? -Please check the Figure 1 caption and font size "management" instead of "managements" -Section 5 and 6. These sections will need to be improved, use Tables to clarify and show the important factors you want to discuss and emphasize .Complete the Information in more detail.
-Section 7.3 should have lines discussing/reasoning of this parameters. Table would help.
- The title of section 7.3.4 need to be checked
Reviewer 3 Report
This article shows a review provides a detailed review of the implementation of constructed wetlands, designs, operation, optimization, and yields and their relationship to plant varieties, substrates, bed depth, hydraulic head, to give a clearer picture of the operation and challenges faced by technology in wastewater management. Although it is an important documentary study, I consider it necessary to make some considerable changes before continuing with its processing and future publication, as described below, I would like to be able to see the document once I have the observations taken care of since I consider it important if you note to give a different perspective and punctual to the future scholar.
The abstract should integrate important findings from the desk research.
From page 35 to 83, although I consider the introduction good, it lacks new references, there are no references, new not even from 2018, 2019, and 2020, where technology has evolved, on the other hand, lines 50 to the 57, mentions that, in the last decade, and references dates much more distant from these than a decade, I consider that the document should review more recent studies, in general, strengthen this part with new references of key studies.
In section 4, lines 116 to 117, in the specific case of Mexico, some studies have used PET, porous stone from rivers, and mixtures of substrates in constructed wetlands that I consider should also be part of these new materials to be unblocked because they are recycled and have a different view of the studies and will give an overview of the use of these substrates in new designs.
From pages 122 to 137, they talk about plant substrate interaction, however, there is no new survey of ornamental plants in wetlands, which are more recent studies, consult these documents that have a description of plants in wetlands and their plant substrate interaction ( Role of wetland plants and use of ornamental flowering plants in constructed wetlands for wastewater treatment: a review.)
Section 6. It requires a definition of emerging countries according to who, and if it is possible to mention them, would be friendlier to the reader. On the other hand, stop when talking about these countries, it would be interesting to see some studies developed in them and analyze the latent need for water treatment in them where the constructed wetlands represent a high alternative for solving wastewater treatment problems.
Jan section 7.1.2. I do not see information on ornamental plants that resist pollutant conditions, as well as mention the future direction of studies in this line of action. Reference must be made in the evolution of larger standing plants that can contribute more notably to the elimination of pollutants. In 266 to 269, review the document of ornamental plants referred to above in the recommendations to give a broader criterion of their use and potential in constructed wetlands (Role of wetland plants and use of ornamental flowering plants in constructed wetlands for wastewater treatment: a review)
Table 1. Does not show used substrates from industrial waste or everyday waste such as PET, tires, etc. Please add recent information on these and discuss them in the document.
Section 10. I believe that it should be called the direction of future research and challenges in terms of the sustainability of CWs.
In this, we will speak specifically about what are the lines of action to follow in future research, what are the challenges that technology faces for its greater acceptance, mainly in regions of the world where its use is required, although it is touched the topic the document shows more areas of opportunity in the topic.
Finally, the conclusion must describe whether the objective set in the study was met promptly.
Regarding references, they should be enriched with more current references.
Reviewer 4 Report
Title: Design, Operation and Optimization of Constructed Wetland for Removal of Pollutant
Dear authors,
The topic of your paper is interesting. Overall, the quality of this paper is adequate. The aim is clear.
However, there is some aspects which should be improved:
1. The references are relevant and appropriate, however, your work presents a relative scarcity of recent references (2015-2020).
2. The research question is clearly outlined. However, I would suggest to argue more why this study may contribute with new knowledge of CWs for removal of pollutants. There are nearly 1000 review articles indexed since 2010 on different aspects / topics of constructed wetland treating wastewater.
Please, clarify why your paper is important.
3. The conclusions chapter is scarce, you should extend more in the conclusions drawn after the extensive bibliographic review work. Highlight some main conclusions and explain them.
There is some aspects which should be improved: suggestions for future research.
It would be interesting some additional managerial implications in line with the findings of the study. Practical implications? Something to inspire future research or implications for practice.
Best regards,
Round 2
Reviewer 1 Report
The paper had a significant improvement after revision, thus, I recommended for publication.
Reviewer 2 Report
This version included suggestions made by reviewers, manuscript has been improved.
Reviewer 3 Report
I consider that the work can be published in its current form, but first you should add a reference from where table 4 is taken on page 438-435, or indicate if it was self-made.